# Open Bandit Dataset and Pipeline: Towards Realistic and Reproducible Off-Policy Evaluation

**Yuta Saito**
Hanjuku-kaso Co., Ltd.
saito@hanjuku-kaso.com

**Shunsuke Aihara**
ZOZO Technologies, Inc.
shunsuke.aihara@zozo.com

**Megumi Matsutani**
ZOZO Technologies, Inc.
megumi.matsutani@zozo.com

**Yusuke Narita**
Yale University
yusuke.narita@yale.edu

## Abstract

*Off-policy evaluation* (OPE) aims to estimate the performance of hypothetical policies using data generated by a different policy. Because of its huge potential impact, there has been growing research interest in OPE. There is, however, no real-world public dataset that enables the evaluation of OPE, making its experimental studies unrealistic and irreproducible. With the goal of enabling realistic and reproducible OPE research, we publicize *Open Bandit Dataset* collected on a large-scale fashion e-commerce platform, ZOZOTOWN. Our dataset is unique in that it contains a set of *multiple* logged bandit feedback datasets collected by running different policies on the same platform. This enables realistic and reproducible experimental comparisons of different OPE estimators for the first time. We also develop Python software called *Open Bandit Pipeline* to streamline and standardize the implementation of batch bandit algorithms and OPE. Our open data and pipeline will contribute to the fair and transparent OPE research and help the community identify fruitful research directions. Finally, we provide extensive benchmark experiments of existing OPE estimators using our data and pipeline. The results open up essential challenges and new avenues for future OPE research.

## 1 Introduction

Interactive bandit systems (e.g., personalized medicine, ad/recommendation/search platforms) produce log data valuable for evaluating and redesigning the system. For example, the logs of a news recommendation system records which news article was presented and whether the user read it, giving the system designer a chance to make its recommendations more relevant. Exploiting log bandit data is, however, more difficult than conventional supervised machine learning: the result is only observed for the action chosen by the system, but not for all the other actions that the system could have taken. The logs are also biased in that they overrepresent the actions favored by the system. A potential solution to this problem is an A/B test that compares the performance of counterfactual systems in an online environment. However, A/B testing counterfactual systems is often difficult because deploying a new policy is time- and money-consuming and entails the risk of failure. This leads us to the problem of *off-policy evaluation* (OPE), which aims to estimate the performance of a counterfactual (or evaluation) policy using only log data collected by a past (or behavior) policy. OPE allows us to compare the performance of candidate counterfactual policies without implementing A/B tests and contributes to safe policy improvements. Its applications range from contextual bandits [2, 16, 17, 23, 28, 30, 31, 32, 36, 39] and reinforcement learning in the web industry [7, 13, 14, 19, 33, 34, 40] to other social domains such as healthcare [22] and education [20].

Preprint. Under review.

**Issues with current experimental procedures.** Although the research community has produced theoretical breakthroughs, the experimental evaluation of OPE remains primitive. Specifically, it lacks a public benchmark dataset for comparing the performance of different methods. Researchers often validate their methods using synthetic simulation environments [14, 19, 36, 38, 40]. A version of the synthetic approach is to modify multiclass classification datasets and treat supervised machine learning methods as bandit policies to evaluate the estimation accuracy of OPE estimators [5, 7, 37, 39]. An obvious problem with these studies is that they are **unrealistic** because there is no guarantee that their simulation environment is similar to real-world settings. To solve this issue, some previous works use proprietary real-world datasets [8, 10, 23, 24]. Because these datasets are not public, however, the results are **irreproducible**, and it remains challenging to compare their methods with new ideas in a fair manner. This contrasts with other domains of machine learning, where large-scale open datasets, such as the ImageNet dataset [4], have been pivotal in driving objective progress [6, 9, 11, 12].

**Contributions.** Our goal is to implement and evaluate OPE in **realistic and reproducible** ways. To this end, we release the *Open Bandit Dataset*, a set of logged bandit feedback datasets collected on the ZOZOTOWN platform.[1] ZOZOTOWN is the largest fashion e-commerce platform in Japan, with an annual gross merchandise value of over 3 billion US dollars. When the platform produced the data, it used Bernoulli Thompson Sampling (Bernoulli TS) [35] and uniform random (Random) policies to recommend fashion items to users. The dataset includes a set of *multiple* logged bandit feedback datasets collected during an A/B test of these bandit policies. Having multiple log datasets is essential because it enables data-driven evaluation of the estimation accuracy of OPE methods as we show in Section 5.

In addition to the dataset, we also implement *Open Bandit Pipeline*, an open-source Python software including a series of modules for implementing dataset preprocessing, policy learning methods, and OPE estimators. Our software provides a complete, standardized experimental procedure for OPE research, ensuring that performance comparisons are fair, transparent, and reproducible. It also enables fast and accurate OPE implementation through a single unified interface, simplifying the practical use of OPE.

Using our dataset and pipeline, we perform an extensive benchmark experiment on existing estimators. Specifically, we implement this OPE experiment by using the log data of one of the policies (e.g., Bernoulli TS) to estimate the policy value of the other policy (e.g., Random) with each OPE estimator. We then assess the accuracy of the estimator by comparing its estimation with the policy value obtained from the data in an *on-policy* manner. This type of data-driven evaluation of OPE is possible with our dataset, because it contains multiple different logged bandit feedback datasets. Our unique real-world dataset thus allows us to conduct the first empirical study comparing a variety of OPE estimators in a realistic and reproducible manner. In the benchmark experiment, we obtain the following observations:

- The estimation performances of all OPE estimators drop significantly when they are applied to estimate the future (or out-sample) performance of a new policy.
- The estimation performances of OPE estimators heavily depend on the experimental settings and hyperparameters.

These empirical findings lead to the following future research directions: (i) improving out-of-distribution estimation performance and (ii) developing methods to identify appropriate OPE estimators with only logged bandit data.

We summarize our key contributions as follows:

- **Dataset Release**: We build and release the *Open Bandit Dataset*, a set of *multiple* logged bandit feedback dataset to assist realistic and reproducible research on OPE (comparison of estimation performance of different OPE estimators).
- **Software Implementation**: We implement *Open Bandit Pipeline*, an open-source Python software that helps practitioners implement OPE to evaluate their bandit systems and researchers compare different OPE estimators in a standardized manner.
- **Benchmark Experiment**: We perform comprehensive benchmark experiments on existing OPE methods and indicate critical challenges in future research.

---

[1]https://corp.zozo.com/en/service/

## 2 Off-Policy Evaluation

**Setup.** We consider a general contextual bandit setting. Let $r \in [0, r_{\max}]$ denote a reward variable (e.g., whether a fashion item as an action results in a click). We let $x \in \mathcal{X}$ be a context vector (e.g., the user's demographic profile) that the decision maker observes when picking an action. Rewards and contexts are sampled from unknown distributions $p(r \mid x, a)$ and $p(x)$, respectively. Let $\mathcal{A}$ be a finite set of actions. We call a function $\pi : \mathcal{X} \to \Delta(\mathcal{A})$ a *policy*. It maps each context $x \in \mathcal{X}$ into a distribution over actions, where $\pi(a \mid x)$ is the probability of taking action $a$ given context $x$.

Let $\mathcal{D} := \{(x_t, a_t, r_t)\}_{t=1}^{T}$ be the historical logged bandit feedback with $T$ rounds of observations. $a_t$ is a discrete variable indicating which action in $\mathcal{A}$ is chosen in round $t$. $r_t$ and $x_t$ denote the reward and the context observed in round $t$, respectively. We assume that a logged bandit feedback is generated by a *behavior policy* $\pi_b$ as $\{(x_t, a_t, r_t)\}_{t=1}^{T} \sim \prod_{t=1}^{T} p(x_t)\pi_b(a_t \mid x_t)p(r_t \mid x_t, a_t)$, where each triplet is sampled independently from the product distribution. We sometimes use $\mathbb{E}_{\mathcal{D}}[f] := |\mathcal{D}|^{-1} \sum_{(x_t, a_t, r_t) \in \mathcal{D}} f(x_t, a_t, r_t)$ to denote the empirical expectation over $\mathcal{D}$.

**Estimation Target and Estimators.** We are interested in using historical logged data to estimate the following *policy value* of any given *evaluation policy* $\pi_e$, which might be different from $\pi_b$:

$$V(\pi_e) := \mathbb{E}_{(x,a,r) \sim p(x)\pi_e(a|x)p(r|x,a)}[r].$$

Estimating $V(\pi_e)$ before implementing $\pi_e$ in an online environment is valuable because $\pi_e$ may perform poorly and damage user satisfaction.

The aim of OPE is to estimate $V(\pi_e)$ using only $\mathcal{D}$ as $V(\pi_e) \approx \hat{V}(\pi_e; \mathcal{D})$ where $\hat{V}$ is an OPE estimator. We define the existing OPE methods such as Direct Method (DM) [1], Inverse Probability Weighting (IPW) [26, 28], Doubly Robust (DR) [5], and some other advanced methods in Appendix B.

## 3 Open-Source Dataset and Pipeline

Motivated by the paucity of real-world datasets and implementations enabling the data-driven evaluation of OPE, we release the following open-source dataset and software.

**Open Bandit Dataset.** Our open-source dataset is a set of *multiple* logged bandit feedback datasets provided by ZOZO, Inc., the largest fashion e-commerce company in Japan. The company uses multi-armed bandit algorithms to recommend fashion items to users in their large-scale fashion e-commerce platform called ZOZOTOWN. We present examples of displayed fashion items in Figure 1. We collected the data in a 7-day experiment in late November 2019 on three "campaigns," corresponding to "ALL", "Men's", and "Women's" items, respectively. Each campaign randomly uses either the Random policy or the Bernoulli TS policy for each user impression.[2] These policies select three of the candidate fashion items for each user. Figure 1 shows that there are three *positions* in our data. We assume that the reward (click indicator) depends only on the item and its position, which is a general assumption on the click generative process used in the web industry [18]. Under this assumption, we can apply the OPE setup in Section 2 to our dataset. We provide some statistics of the dataset in Table 1. The dataset is large and contains many millions of recommendation instances. Each row of the data has feature vectors such as age, gender, and past click history of the users. These feature vectors are hashed, thus the dataset does not contain any personally identifiable information. Moreover, the dataset includes some item-related features such as price, fashion brand, and item categories. It also includes the probability that item $a$ is displayed at each position by the data collection policies which are used to calculate the importance weight. We share the full version of our dataset at **https://research.zozo.com/data.html**.[3] Small-sized example data are also available at **https://github.com/st-tech/zr-obp/tree/master/obd**.

To our knowledge, our open-source dataset is the first to include logged bandit datasets collected by running *multiple* different policies and the exact policy implementations used in real production, enabling "***realistic and reproducible evaluation of OPE***" for the first time.

---

[2]Note that we pre-trained Bernoulli TS for over a month before the data collection process, and the policy well converges to a fixed one. Therefore, our dataset fits the standard OPE formulation, that assumes fixed behavior and evaluation policies.

[3]The dataset is licensed under CC BY 4.0.

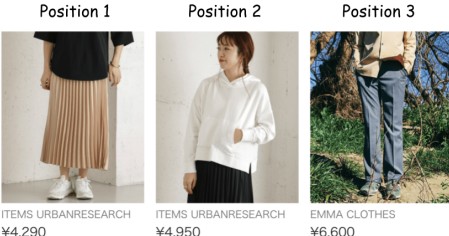

Figure 1: Fashion items as actions displayed in ZOZOTOWN recommendation interface.

Table 1: Statistics of the Open Bandit Dataset

| Campaigns | Data Collection Policies | #Data | #Items | #Dim | CTR ($V(\pi)$) $\pm$95% CI | Relative-CTR |
|---|---|---|---|---|---|---|
| **ALL** | **Random** | 1,374,327 | 80 | 84 | 0.35% $\pm$0.010 | 1.00 |
| | **Bernoulli TS** | 12,168,084 | | | 0.50% $\pm$0.004 | 1.43 |
| **Men's** | **Random** | 452,949 | 34 | 38 | 0.51% $\pm$0.021 | 1.48 |
| | **Bernoulli TS** | 4,077,727 | | | 0.67% $\pm$0.008 | 1.94 |
| **Women's** | **Random** | 864,585 | 46 | 50 | 0.48% $\pm$0.014 | 1.39 |
| | **Bernoulli TS** | 7,765,497 | | | 0.64% $\pm$0.056 | 1.84 |

*Note*: Bernoulli TS stands for Bernoulli Thompson Sampling. **#Data** is the total number of user impressions observed during the 7-day experiment. **#Items** is the total number of items having a non-zero probability of being recommended by each policy. **#Dim** is the number of dimensions of the raw context vectors. **CTR** is the percentage of a click being observed in the log data, and this is the performance of the data collection policies in each campaign. The 95% confidence interval (CI) of CTR is calculated based on a normal approximation of the Bernoulli sampling. **Relative-CTR** is the CTR relative to that of the Random policy for the "ALL" campaign.

**Open Bandit Pipeline.** To facilitate the use of OPE in practice and standardize its experimental procedures, we also build a Python package called *Open Bandit Pipeline*. Our pipeline contains the following main modules:

- The **dataset** module provides a data loader to preprocess the Open Bandit Dataset and tools to generate synthetic bandit datasets. It also implements a class to handle multiclass classification datasets as bandit feedback, which is useful when we conduct OPE experiments in research papers.

- The **policy** module implements several online bandit algorithms and off-policy learning methods such as the one maximizing the IPW objective with only logged bandit data. This module also implements interfaces that allow practitioners to easily evaluate their own policies in their business using OPE.

- The **ope** module implements several existing OPE estimators including the basic ones such as DM, IPW, and DR and some advanced ones such as Switch [39], More Robust Doubly Robust (MRDR) [7], and DR with Optimistic Shrinkage (DRos) [29]. This module also implements interfaces to implement new estimators so that researchers can test their own estimation methods with our pipeline easily.

Appendix E and examples at **https://github.com/st-tech/zr-obp/tree/master/examples/quickstart** describe the basic usage of the pipeline. We also provide the thorough documentation so that anyone can follow its usage.[4] This pipeline allows researchers to focus on building their OPE estimator and to easily compare it with other methods in realistic and reproducible ways.

Every core function of the packages is tested[5] and thus are well maintained. The package currently has five core contributors[6]. The active development and maintenance will continue in a long period.

---

[4]https://zr-obp.readthedocs.io/en/latest/
[5]https://github.com/st-tech/zr-obp/tree/master/tests
[6]https://github.com/st-tech/zr-obp/graphs/contributors

Table 2: Comparison of currently available large-scale bandit datasets

| | Criteo Data [15] | Yahoo! R6A&B [16] | Open Bandit Dataset (ours) |
|---|---|---|---|
| **Domain** | Display Advertising | News Recommendation | Fashion E-Commerce |
| **Dataset Size** | >103M | >40M | >26M (will increase) |
| **#Data Collection Policies** | 1 | 1 | **2 (will increase)** |
| **Uniform Random Data** | ✗ | ✔ | ✔ |
| **Data Collection Policy Code** | ✗ | ✗ | ✔ |
| **Evaluation of Bandit Algorithms** | ✔ | ✔ | ✔ |
| **Evaluation of OPE** | ✗ | ✗ | ✔ |
| **Pipeline Implementation** | ✗ | ✗ | ✔ |

*Note*: **Dataset Size** is the total number of samples included in the whole dataset. **#Data Collection Policies** is the number of policies that were used to collect the data. **Uniform Random Data** indicates whether the dataset contains a subset of data generated by the uniform random policy. **Data Collection Policy Code** indicates whether the code to replicate data collection policies is publicized. **Evaluation of Bandit Algorithms** indicates whether it is possible to use the data to evaluate bandit algorithms. **Evaluation of OPE** indicates whether it is possible to use the dataset to evaluate OPE estimators. **Pipeline Implementation** indicates whether a pipeline tool to handle the dataset is available.

Table 3: Comparison of currently available packages of bandit algorithms and OPE

| | contextualbandits [3] | RecoGym [27] | Open Bandit Pipeline (ours) |
|---|---|---|---|
| **Synthetic Data Generator** | ✗ | ✔ | ✔ |
| **Classification Data Handler** | ✗ | ✗ | ✔ |
| **Support for Real-World Data** | ✗ | ✗ | ✔ |
| **Bandit Algorithms** | ✔ | ✔ | ✔ |
| **Basic OPE Estimators** | ✔ | ✗ | ✔ |
| **Advanced OPE Estimators** | ✗ | ✗ | ✔ |
| **Evaluation of OPE** | ✗ | ✗ | ✔ |

*Note*: **Synthetic Data Generator** indicates whether it is possible to create synthetic bandit data with the package. **Classification Data Handler** indicates whether it is possible to transform multiclass classification data to bandit feedback with the package. **Support for Real-World Data** indicates whether it is possible to handle real-world bandit data with the package. **Bandit Algorithms** indicates whether the package includes implementations of online and offline bandit algorithms. **Basic OPE Estimators** indicates whether the package includes implementations of *basic* OPE estimators such as DM, IPW, and DR described in Appendix B. **Advanced OPE Estimators** indicates whether the package includes implementations of *advanced* OPE estimators such as Switch and More Robust Doubly Robust described in Appendix B. **Evaluation of OPE** indicates whether it is possible to evaluate the accuracy of OPE estimators with the package.

# 4 Related Resources

Here, we summarize the existing related datasets and packages, and clarify the advantages of ours.

**Related Datasets.**  Our dataset is closely related to those of [15] and [16]. Lefortier et al. [15] introduces a large-scale logged bandit feedback data (Criteo Data[7]) from a leading company in display advertising, Criteo. The data contain context vectors of user impressions, advertisements (ads) as actions, and click indicators as rewards. It also provides the ex-ante probability of each ad being selected by the behavior policy. Therefore, this dataset can be used to compare different off-policy *learning* methods, which aim to learn a new policy using only historical logged bandit data. In contrast, Li et al. [16] introduces a dataset (Yahoo! R6A&B[8]) collected on a news recommendation interface of the Yahoo! Today Module. The data contain context vectors of user impressions, presented news as actions, and click indicators as rewards. The data were collected by running a uniform random policy on the news recommendation platform, allowing researchers to evaluate their own bandit algorithms.

---

[7]https://www.cs.cornell.edu/ adith/Criteo/
[8]https://webscope.sandbox.yahoo.com/catalog.php?datatype=r

However, the existing datasets have several limitations, which we overcome as follows:

- They include only a single logged bandit feedback dataset collected by running only a single policy. Moreover, the previous datasets do not provide the implementation to replicate the policies used during data collection. As a result, these datasets cannot be used for the comparison of different OPE estimators, although they can be used to evaluate off-policy learning methods.

  → In contrast, we provide the code to replicate the data collection policies (i.e., Bernoulli TS and Random) in our pipeline, which allows researchers to rerun the same policies on the log data. Moreover, our open dataset consists of a set of *multiple* different logged bandit feedback datasets generated by running two different policies on the same platform. It enables the comparison of different OPE estimators, as we show in Section 5. ***This is the first large-scale bandit dataset, enabling a realistic and data-driven evaluation of OPE***.

- The previous datasets do not provide a pipeline implementation to handle their data. Researchers have to reimplement the experimental environment by themselves before implementing their own OPE methods. This may lead to inconsistent experimental conditions across different studies, potentially causing reproducibility issues.

  → We implement Open Bandit Pipeline to simplify and standardize the experimental processing of bandit algorithms and OPE. This tool thus contributes to the reproducible and transparent use of our dataset.

Table 2 summarizes the key differences between our dataset and the existing ones.

**Related Packages.** There are several existing packages related to Open Bandit Pipeline. The *contextualbandits* package[9] contains implementations of several contextual bandit algorithms [3]. It aims to provide an easy procedure to compare bandit algorithms to reproduce research papers that do not provide easily available implementations. There is also *RecoGym*[10] that focuses on providing simulation bandit environments imitating the e-commerce recommendation setting [27].

However, the following features differentiate our pipeline from the previous ones:

- The previous packages focus on implementing and comparing online bandit algorithms or off-policy learning methods. However, they ***cannot*** be used to implement several advanced OPE estimators and the evaluation of OPE procedures.

  → Our package implements a wide variety of OPE estimators, including advanced ones such as Switch, MRDR, and DRos. Our package also provides flexible interfaces for implementing new OPE estimators. Consequently, researchers can easily compare their own estimators with other methods in a fair, standardized manner.

- The previous packages accept their own interface and data formats; they are not user-friendly.

  → Our package follows the prevalent *scikit-learn* style interface and provides sufficient example codes at **https://github.com/st-tech/zr-obp/tree/master/examples** so that anyone, including practitioners and students, can follow the usage easily.

- The previous packages cannot handle real-world bandit datasets.

  → Our package comes with the Open Bandit Dataset and includes the **dataset module**. This enables the evaluation of bandit algorithms and OPE estimators using real-world data. This function of our package contributes to realistic experiments on these topics.

Table 3 summarizes the key differences between our pipeline and the existing ones.

## 5 Benchmark Experiments

We perform benchmark experiments of OPE estimators using the Open Bandit Dataset and Pipeline. We first describe an experimental protocol to evaluate OPE estimators and use it to compare a wide variety of existing estimators. We then discuss our initial findings in the experiments and indicate future research directions. We share the code to replicate the benchmark experiments at **https://github.com/st-tech/zr-obp/tree/master/benchmark**.

---

[9]https://github.com/david-cortes/contextualbandits
[10]https://github.com/criteo-research/reco-gym

Table 4: Comparison of relative-estimation errors of OPE estimators (**ALL Campaign**)

| OPE Estimators | Random → Bernoulli TS | | Bernoulli TS → Random | |
|---|---|---|---|---|
| | *in*-sample | *out*-sample | *in*-sample | *out*-sample |
| **DM** | **0.23433**$^\diamond$ ±0.02131 | **0.25730**$^\diamond$ ±0.02191 | **0.34522**$^\diamond$ ±0.01020 | **0.29422**$^\diamond$ ±0.01199 |
| **IPW** | **0.05146**$^*$ ±0.03418 | 0.09169 ±0.04086 | **0.02341**$^*$ ±0.02146 | 0.08255 ±0.03798 |
| **SNIPW** | **0.05141**$^\dagger$ ±0.03374 | **0.08899**$^\dagger$ ±0.04106 | 0.05233 ±0.02614 | 0.13374 ±0.04416 |
| **DR** | 0.05269 ±0.03460 | 0.09064 ±0.04105 | 0.06446 ±0.03001 | 0.14907 ±0.05097 |
| **SNDR** | 0.05269 ±0.03398 | **0.09013**$^*$ ±0.04122 | 0.04938 ±0.02645 | 0.12306 ±0.04481 |
| **Switch-DR** ($\tau = 5$) | 0.15350 ±0.02274 | 0.16918 ±0.02231 | 0.26811 ±0.00780 | 0.21945 ±0.00944 |
| **Switch-DR** ($\tau = 10$) | 0.09932 ±0.02459 | 0.12051 ±0.02203 | 0.21596 ±0.00907 | 0.16532 ±0.01127 |
| **Switch-DR** ($\tau = 50$) | 0.05269 ±0.03460 | 0.09064 ±0.04105 | 0.09769 ±0.01515 | **0.04019**$^*$ ±0.01349 |
| **Switch-DR** ($\tau = 100$) | 0.05269 ±0.03460 | 0.09064 ±0.04105 | 0.05938 ±0.01597 | **0.01310**$^\dagger$ ±0.00988 |
| **Switch-DR** ($\tau = 500$) | 0.05269 ±0.03460 | 0.09064 ±0.04105 | **0.02123**$^\dagger$ ±0.01386 | 0.06564 ±0.02132 |
| **Switch-DR** ($\tau = 1000$) | 0.05269 ±0.03460 | 0.09064 ±0.04105 | 0.02840 ±0.01929 | 0.05347 ±0.03330 |
| **DRos** ($\lambda = 5$) | 0.19135 ±0.01964 | 0.21240 ±0.01938 | 0.30395 ±0.00726 | 0.25216 ±0.00929 |
| **DRos** ($\lambda = 10$) | 0.17400 ±0.01993 | 0.19500 ±0.01885 | 0.28735 ±0.00706 | 0.23627 ±0.00899 |
| **DRos** ($\lambda = 50$) | 0.12867 ±0.02124 | 0.15155 ±0.01911 | 0.23876 ±0.00707 | 0.18855 ±0.00907 |
| **DRos** ($\lambda = 100$) | 0.11055 ±0.02241 | 0.13561 ±0.02080 | 0.21550 ±0.00744 | 0.16474 ±0.00942 |
| **DRos** ($\lambda = 500$) | 0.07715 ±0.02736 | 0.10915 ±0.02944 | 0.16055 ±0.00942 | 0.10601 ±0.01048 |
| **DRos** ($\lambda = 1000$) | 0.06739 ±0.02988 | 0.10187 ±0.03358 | 0.13717 ±0.01064 | 0.08034 ±0.01093 |
| **MRDR** | 0.05458 ±0.03386 | 0.09232 ±0.04169 | 0.02511 ±0.01735 | 0.08768 ±0.03821 |

*Note*: The averaged relative-estimation errors and their unbiased standard deviations estimated over 30 different bootstrapped iterations are reported. We describe the method to estimate the standard deviations in Appendix C. $\pi_b \to \pi_e$ represents the OPE situation where the estimators aim to estimate the policy value of $\pi_e$ using logged bandit data collected by $\pi_b$. The **red**$^\dagger$ and **green**$^*$ fonts represent the best and second-best estimators, respectively. The **blue**$^\diamond$ fonts represent the worst estimator for each setting.

## 5.1 Experimental Protocol

We can empirically evaluate OPE estimators' performance by using two sources of logged bandit feedback collected by running two different policies. In the protocol, we regard one policy as behavior policy $\pi_b$ and the other one as evaluation policy $\pi_e$. We denote log data generated by $\pi_b$ and $\pi_e$ as $\mathcal{D}^{(b)} := \{(x_t^{(b)}, a_t^{(b)}, r_t^{(b)})\}_{t=1}^{T^{(b)}}$ and $\mathcal{D}^{(e)} := \{(x_t^{(e)}, a_t^{(e)}, r_t^{(e)})\}_{t=1}^{T^{(e)}}$. Then, by applying the following protocol to several different OPE estimators, we compare their estimation performances:

1. Define the evaluation and test sets as: (***in*-sample case**) $\mathcal{D}_{\mathrm{ev}} := \mathcal{D}_{1:T^{(b)}}^{(b)}$, $\mathcal{D}_{\mathrm{te}} := \mathcal{D}_{1:T^{(e)}}^{(e)}$, (***out*-sample case**) $\mathcal{D}_{\mathrm{ev}} := \mathcal{D}_{1:\tilde{t}}^{(b)}$, $\mathcal{D}_{\mathrm{te}} := \mathcal{D}_{\tilde{t}+1:T^{(e)}}^{(e)}$, where $\mathcal{D}_{a:b} := \{(x_t, a_t, r_t)\}_{t=a}^{b}$ and $\tilde{t}$ is the time-series split-point.

2. Estimate the policy value of $\pi_e$ using $\mathcal{D}_{\mathrm{ev}}$ by an OPE estimator $\hat{V}$. We represent a policy value estimated by $\hat{V}$ as $\hat{V}(\pi_e; \mathcal{D}_{\mathrm{ev}})$.

3. Estimate $V(\pi_e)$ by the *on-policy estimation* and regard it as the policy value of $\pi_e$, i.e., $V_{\mathrm{on}}(\pi_e) := \mathbb{E}_{\mathcal{D}_{\mathrm{te}}}[r_t^{(e)}]$.

4. Compare the off-policy estimate $\hat{V}(\pi_e; \mathcal{D}_{\mathrm{ev}})$ with its on-policy counterpart $V_{\mathrm{on}}(\pi_e)$. We can evaluate the estimation accuracy of $\hat{V}$ using the following *relative estimation error* (relative-EE):

$$relative\text{-}EE(\hat{V}; \mathcal{D}_{\mathrm{ev}}) := |\hat{V}(\pi_e; \mathcal{D}_{\mathrm{ev}}) - V_{\mathrm{on}}(\pi_e)| / V_{\mathrm{on}}(\pi_e).$$

5. To estimate the standard deviation of relative-EE, repeat the above process several times with different bootstrap samples of the logged bandit data.

We call the problem setting **without** the sample splitting by time series as the *in*-sample case. In contrast, we call the setting **with** the sample splitting as the *out*-sample case, where OPE estimators

aim to estimate the policy value of an evaluation policy in the future data. The standard OPE assumes the *in*-sample case where there are no distributional change in the environment over time. However, in practice, we aim to estimate the performance of an evaluation policy in the future, which may introduce the distributional change between the data used to conduct OPE and the environment that defines the policy value of the evaluation policy. We thus test the performance of OPE estimators in the *out*-sample case in addition to the *in*-sample case.

## 5.2 Compared Estimators

We use our protocol and compare the following OPE estimators: DM, IPW, Self-Normalized Inverse Probability Weighting (SNIPW), DR, Self-Normalized Doubly Robust (SNDR), Switch Doubly Robust (Switch-DR), DRos, and MRDR. We define and describe these estimators in Appendix B. We test different hyperparameter values for Switch-DR and DRos.These above estimators have not yet been compared in a large, real-world setting.

For estimators except for DM, we use the true action choice probability $\pi_b(a|x)$ contained in the Open Bandit Dataset. For estimators except for IPW and SNIPW, we need to obtain a reward estimator $\hat{q}$. We do this by using logistic regression (implemented in *scikit-learn* [25]) and training it using 30% of $\mathcal{D}_{\mathrm{ev}}$. We then use the rest of the data to estimate the policy value of an evaluation policy.

## 5.3 Results and Discussion

The results of the benchmark experiments on the "ALL" campaign are given in Table 4. (See Appendix C for additional results.) We describe **Random** $\to$ **Bernoulli TS** to represent the OPE situation where we use Bernoulli TS as $\pi_e$ and Random as $\pi_b$. Similarly, we use **Bernoulli TS** $\to$ **Random** to represent the situation where we use Random as $\pi_e$ and Bernoulli TS as $\pi_b$.

**Performance comparisons.**   First, DM fails to estimate the policy values in all settings due to the bias of the reward estimator. We observe that the reward estimator does not improve upon a naive estimation using the mean CTR for every estimation in the binary cross-entropy measure. (We present the performance of the reward estimator in Appendix C.) The problem with DM leads us to expect that the other estimators may perform better because they do not rely on the correct specification of the reward estimator. We confirm this expectation in Table 4, where one can see that the others drastically outperform DM. Among the other estimators, IPW, SNIPW, and MRDR exhibit stable estimation performances across different settings, and thus we can use these estimators safely. In **Bernoulli TS** $\to$ **Random**, Switch-DR performs the best with a proper hyperparameter configuration. Its performance, however, largely depends on the choice of hyperparameters, as we discuss later in detail. Note here that the performances of Switch-DR with some large hyperparameters are the same as those of DR. This is a natural observation, as their definitions are the same when the importance weights of all samples are lower than a given hyperparameter. In summary, we observe that simple estimators such as IPW and SNIPW perform better in many cases than Switch-DR and DRos even though these advanced methods performed well on synthetic experiments in previous studies. This suggests that evaluating the performance of OPE methods on synthetic or classification datasets may produce impractical conclusions about the estimators' empirical properties. In contrast, our dataset enables researchers to produce more practical conclusions about OPE methods.

**Out-sample generalization of OPE.**   Next, we compare the estimation accuracy of each estimator between the *in*-sample and *out*-sample situations. Table 4 shows that the estimators' performances drop significantly in almost all situations when they attempt to generalize their OPE results to the out-sample or future data. The result suggests that the current OPE methods may fail to evaluate the performance of a new policy in the future environment, as they implicitly assume that the data generating distribution does not change over time. Moreover, this kind of realistic out-of-distribution generalization check of OPE cannot be conducted on synthetic or multiclass classification datasets. We thus expect that the Open Bandit Dataset promotes future research about the robustness of OPE methods to distributional changes.

**Performance changes across different settings.**   Finally, we compare the estimation accuracy of each estimator under different experimental conditions and with different hyperparameters. We observe in Table 4 that the estimators' performance can change significantly depending on the

Table 5: Comparison of OPE performance with different reward estimators

| OPE Estimators | Logistic Regression | Random Forest |
|:---:|:---:|:---:|
| **DM** | 0.34522 ±0.01020 | 0.30391 ±0.01059 |
| **DR** | 0.06446 ±0.03001 | 0.05775 ±0.02600 |
| **SNDR** | 0.04938 ±0.02645 | 0.04658 ±0.02155 |
| **Switch-DR** ($\tau = 100$) | 0.05938 ±0.01597 | 0.05499 ±0.01425 |
| **DRos** ($\lambda = 100$) | 0.21550 ±0.00744 | 0.19111 ±0.00781 |
| **MRDR** | 0.02511 ±0.01735 | 0.03000 ±0.02592 |

*Note*: This is the result in the case of **ALL Campaign/Bernoulli TS → Random** and **in-sample**. The averaged relative-estimation errors over 30 different bootstrapped iterations are reported. The results on the other settings are in Appendix C.

experimental conditions. In particular, we tested several values for the hyperparameter $\tau$ of Switch-DR. We observe that its estimation performance largely depends on the choice of $\tau$. It is obvious that Switch-DR is significantly better with large values of $\tau$ on our data. We also investigate the effect of the choice of the machine learning method to construct the reward estimator in Table 5. Specifically, we additionally test the estimators' performance when random forest is used. The table shows that using random forest to construct the reward estimator provides a more accurate OPE on our dataset. These observations suggest that practitioners have to choose an appropriate OPE estimator or tune the estimators' hyperparameters carefully for their specific application. It is thus necessary to develop a reliable method to choose and tune OPE estimators in a data-driven manner. Specifically, in many cases, we have to tune the estimators' hyperparameters, including the reward estimator, without the ground-truth policy value of the evaluation policy.

## 6 Conclusion, Future Work, and Limitations

To enable a realistic and reproducible evaluation of off-policy evaluation, we publicized the Open Bandit Dataset–a set of benchmark logged bandit datasets collected on a large-scale fashion e-commerce platform. The dataset comes with Open Bandit Pipeline, Python software that makes it easy to evaluate and compare different OPE estimators. We expect them to facilitate understanding of the empirical properties of OPE techniques and address experimental inconsistencies in the literature. In addition to building the data and pipeline, we performed extensive benchmark experiments on OPE. Our experiments highlight that the current OPE methods are inaccurate for estimating the out-of-distribution performance of a new policy. It is also evident that it is necessary to develop a data-driven method to select an appropriate estimator for each given environment.

One limitation of the dataset is that data collection is done by using only two behavior policies. Here, we emphasize that there had never been any public real-world data that allow realistic and reproducible OPE research before. Our open-source is an initial step towards the goal. Having many data collection policies would be even more valuable, but releasing data with two different logging policies is distinguishable enough from the prior work. We continue to work with the platform to extend our data, which will hopefully result in more data about additional business domains, features, and most importantly, behavior policies. We believe that our work will inspire other researchers and companies to create follow-up benchmark datasets to advance OPE research further.

Another limitation is that we assume that the reward of an item at a position does not depend on other simultaneously presented items. This assumption might not hold, as an item's attractiveness can have a significant effect on the expected reward of another item in the same recommendation list [18]. To address more realistic situations, we have implemented some OPE estimators for the slate action setting [21, 32] in Open Bandit Pipeline.[11] Comparing the standard OPE estimators and those for the slate action setting in our data is an interesting future research direction.

---

[11]https://github.com/st-tech/zr-obp/blob/master/obp/ope/estimators_slate.py

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
