# OpenReview forum: "Open Bandit Dataset and Pipeline: Towards Realistic and Reproducible Off-Policy Evaluation"
_NeurIPS.cc/2021/Track/Datasets_and_Benchmarks/Round1 — Submitted to NeurIPS 2021 Datasets and Benchmarks Track (Round 1)_

### Official Review · Reviewer_Eeqo · 2021-06-30
**Outstanding Benchmark**

**Rating:** 8
**Confidence:** 5
**Correctness:** Seems correct to me.
**Clarity:** Very well written.

**Strengths:**

In terms of (public) benchmarking for Bandit OPE, this work is the first of its kind.  No other work has the combination of realistic and reproducible OPE benchmarking.  This is made possible by ZOZOTOWN agreeing to run two separate policies (Thompson Sampling and Uniform) in a A/B test to collect the log data.  I also appreciate the policy code being released, as it also helps with reproducibility.  The design of this benchmark is elegant.



**Weaknesses:**

I have one non-trivial issue with the interpretation of the benchmark evaluation.  The authors claim that out-of-sample generalization fails for OPE (penultimate paragraph in page 8).  However, looking at the results in Table 4, I see more of a mixed bad (sometimes it's fine, and sometimes it improves, and sometimes it gets worse).  Can the authors comment further?

Minor issues:
-- I think the claim of "will increase" in Table 2 is slightly problematic.  I have seen papers promise these things and then fail to deliver.  I would remove such speculative wording.

-- Some other challenges/competitions might be relevant and worth citing:
http://explochallenge.inria.fr/index.html


**Additional Feedback:**

Great paper.

**Documentation:**

Seems pretty well documented.   I also took a look at the online resources, and those seem well organized as well.

**Ethics:**

One potential concern is privacy, and the authors seem to have sufficiently addressed this issue (by hashing user-specific features).

Another concern is having practitioners become too confident in their method based on this benchmark, but the authors clearly addressed the limitations of their benchmark.

**Relation To Prior Work:**

The discussion with prior work is very nuanced and well appreciated.

**Summary And Contributions:**

This work proposes the Open Bandit Dataset and Benchmark.  The main goal of this work is to create a realistic benchmark for reproducible testing of bandit OPE methods.  To do so, the authors collected data from ZOZOTOWN, using both Thompson Sampling and Uniform Sampling as the logging policies.  This means that one can both evaluate the accuracy of a standard OPE pipeline (e.g., using logged data from TS to estimate the reward of Uniform) in an on-policy fashion (via the logged data from Uniform).

In their benchmarking, the authors make two research findings:
-- The estimation performances of all OPE estimators drop significantly when they are applied to estimate the future (or out-sample) performance of a new policy.
-- The estimation performances of OPE estimators heavily depend on the experimental settings and hyperparameters.
(I have personally always suspected the latter point, but it's great to see it validated on a realistic dataset.)

The final contribution of this work is an easy-to-use software package, the Open Bandit Pipeline, for efficiently running more experiments and tests.

---

> ### Author Response · Authors · 2021-07-14
> **Author response**
>
> Thank you for your useful, detailed feedback. We appreciate that you found our
> open-source dataset, software, and empirical studies to be important to advance OPE research. We will update the paper with the suggested minor revisions and respond below to some concrete questions and comments.
>
> > The authors claim that out-of-sample generalization fails for OPE (penultimate paragraph in page 8). However, looking at the results in Table 4, I see more of a mixed bad (sometimes it's fine, and sometimes it improves, and sometimes it gets worse). Can the authors comment further?
>
> Thank you for pointing this out. Please look at the estimators having reasonable estimation performance and less hyperparameters such as IPW, SNIPW, DR, SNDR, and MRDR (Table 4). For these estimators, we can see that they consistently perform worse in the out-sample case, compared to the in-sample case. This observation generalizes across different campaigns (see also Tables 7 and 8 in the appendix). Based on this result, we argue that out-of-sample generalization fails for OPE.
>
> As you pointed out, we see more of a mixed results for Switch-DR and DRos (especially Switch-DR). We attribute the results to the hyperparameter choice. We tried a wide range of hyperparameter values for these estimators ($\tau$ and $\lambda$), and we found that their performance is highly sensitive to the hyperparameter choice (we reported this result in “Performance changes across different settings” in Section 5.3). In this process, it is possible that some hyperparameters arbitrarily favor the out-sample case. However, it is difficult to tune these hyperparameters using only logged bandit data, and we cannot expect that the "in-sample vs out-sample" result of Switch-DR and DRos generalizes across different settings and datasets.
>
> We will clarify and give additional explanations about this point in the main text.
>
> > Minor issues: -- I think the claim of "will increase" in Table 2 is slightly problematic. I have seen papers promise these things and then fail to deliver. I would remove such speculative wording.
>
> Thank you for the advice. We will remove this wording from the table in the revision.

---

### Official Review · Reviewer_9HcS · 2021-07-04

**Rating:** 5
**Confidence:** 4

**Strengths:**

+ A large real-world dataset, with associated code that makes it straightforward to compare different OPE methods in the contextual bandit setting.
+ Very comprehensive comparison to existing benchmarks

**Weaknesses:**

The biggest issue is that conclusions about performance between OPE estimators are drawn, seemingly by comparing means rather than using statistical significance. The procedure involves evaluation OPE estimators, across 30 different bootstrap resamples. (This is presumably to evaluate the variability of the OPE estimator algorithm, which is sensible). The mean of performance across these 30 estimators is used. Then there are statements about one OPE estimator being better than another, without saying how we can be confident that is the case.

One other point is that it is not clearly stated that the goal is to evaluate performance of the OPE algorithm, across possibly different training sets. Another option would have been to simply report the performance of the OPE estimator on the training set, and consider instead performance with increasing samples. It would be useful to state the explicit goal of the pipeline, and what it is testing.

It is a bit strong to claim that previous datasets cannot be used to evaluate OPE estimators. With a large amount of data, the basic estimators which use sample averages and IS ratios are consistent and should converge to the true value, assuming reasonable coverage. Now, I can see the argument that having multiple policies means that you can leverage on-policy estimates to get likely more accurate value estimates, with less data. Nonetheless, even the on-policy estimator is not truly the ground truth. This point, and the claim that previous datasets cannot be used, should be explained. Or, instead of making such a strong claim, you could simply point out that your dataset facilitates such comparisons.

**Additional Feedback:**

“Although the research community has produced theoretical breakthroughs, the experimental evaluation of OPE remains primitive. Specifically, it lacks a public benchmark dataset for comparing the performance of different methods. ” —> This is a false comparison. Empirical evaluation does not require public benchmark datasets to be advanced or insightful. You may have heard the statement: “My number is bigger than your number is the lowest form of science”. It may be better to avoid stating that empirical evaluation is currently primitive, and focus rather on what forms of evaluation this dataset can facilitate.
I should add that the discussion right after this highlights why a public dataset based on real-world data could be beneficial. The topic sentence here could be focused on that.

“This contrasts with other domains of machine learning, where large-scale open datasets, such as the ImageNet dataset [4], have been pivotal in driving objective progress [6, 9, 11, 12]. ” —> Benchmark datasets can also hold a field back, since they allow for overfitting and a focus on just getting that number a little bit higher. You should address this issue with benchmark datasets.

“Note that we pre-trained Bernoulli TS for over a month before the data collection process, and the policy well converges to a fixed one. ” —> I am not sure what this means. If it converges, does that mean it is optimal and done learning?
Further, why is it needed to converge to a fixed policy? You can get a fixed policy simply by letting it train for some number of steps and then fixing it.

“In contrast, we provide the code to replicate the data collection policies (i.e., Bernoulli TS and Random) in our pipeline, which allows researchers to rerun the same policies on the log data. ” —> This is mentioned as a contribution. I do not understand why this is useful.

Minor statements:
1. “more practical conclusions about OPE methods.” What is a practical conclusion? This terminology is poor.
2. For this benchmark, you could consider a useful way to visualize these tables.

**Clarity:**

The writing is clear.

There are a few parts that could use explanation earlier. There is an important point that having multiple policies enables evaluation of OPE methods. This was not obvious to me, until I read Section 5 (though I know it was stated earlier in the paper, by this point it was not immediately in my mind). This point should be explained, at least briefly, when this contribution is mentioned: “with multiple policies, one can be used as the behavior and the other as target, allowing for on-policy value estimate which is much lower variance and so provides a reasonably ground truth value.”

Some paragraphs are too long, and should be broken up into more manageable paragraphs with one key idea. One example is the paragraph with Bold title “Performance Comparisons”.

**Correctness:**

As mentioned above, the protocol does not seem to include a strategy to state, with confidence, that one estimator is better than another. Presumably, the dataset is large and so the mean is informative. But, this is not stated nor addressed.

One minor point: why is there a time series split point for evaluation and test? Are the two policies run simultaneously? I would think that, even in the out of sample case, the full dataset could be used for the on-policy estimator.

For Step 5 in Section 5.1, do you recalculate the true on-policy estimate? The text suggests so, but this would not be sensible. Rather, Vhat would be computed on different bootstrap samples. (Note: the pseudocode in the appendix I looked at later seems to have this correct, so just the main text needs to be fixed).


**Documentation:**

Well documented.


**Ethics:**

No ethical concerns.


**Relation To Prior Work:**

The paper goes into depth on comparisons to related benchmarks, with very clear tables contrasting what is currently available and what is new in their benchmark.

**Summary And Contributions:**

The goal is to evaluate OPE methods on real-world datasets. This dataset includes two logged policies—with the plan to include more—which facilitates evaluating the quality of OPE methods.

---

> ### Author Response · Authors · 2021-07-14
> **Author response**
>
> Thank you for your useful, detailed feedback. We will update the paper with the suggested minor revisions and respond below to some concrete questions and comments.
>
> ---
> > The biggest issue is that conclusions about performance between OPE estimators are drawn, seemingly by comparing means rather than using statistical significance. [...] Then there are statements about one OPE estimator being better than another, without saying how we can be confident that is the case.
>
>
> Thank you for pointing this out. In the main text, we do not say one estimator is the best or simply rank the performance of the estimators. We instead pay attention to more obvious observations such as the failure of DM and the effect of hyperparameter choice on the estimators' performance.
>
> However, we agree that the colors in Tables 4, 7, and 8 are confusing and based only on the means of the performance (another reviewer also pointed this out). We will remove the colors in the tables and add some comments on the confidence of each observation, as you suggested.
>
>
> ---
> > It is a bit strong to claim that previous datasets cannot be used to evaluate OPE estimators. With a large amount of data, the basic estimators which use sample averages and IS ratios are consistent and should converge to the true value, assuming reasonable coverage. Now, I can see the argument that [...] , you could simply point out that your dataset facilitates such comparisons.
>
> First of all, please make sure that our aim is to **evaluate and compare the estimation performance of OPE methods (in a realistic and reproducible manner) , not to benchmark bandit policies using OPE**. As we described and demonstrated in Section 5, having multiple policies is essential to enable the comparison of the estimation accuracy of OPE estimators, not to leverage on-policy estimates to ensure an accurate OPE.
>
> Thus, we still argue that the previous datasets (such as those in Section 4) cannot be used for the evaluation of OPE estimators. This is because the existing public bandit datasets have only one set of logged bandit data. As you suggested, these datasets can be used to evaluate the performance of bandit policies such as UCB by leveraging the consistency of OPE. However, these datasets cannot be used to evaluate the accuracy of OPE estimators (this is our goal). In this sense, we believe that our dataset and code release enables such evaluation of OPE experiments for the first time.
>
> Indeed, the other reviewers acknowledge that our dataset is the only dataset that enables the OPE experiments. **Please see the comments below made by Reviewer Eeqo (who gives us the score “8”)**
>
> > In terms of (public) benchmarking for Bandit OPE, this work is the first of its kind. **No other work has the combination of realistic and reproducible OPE benchmarking. This is made possible by ZOZOTOWN agreeing to run two separate policies (Thompson Sampling and Uniform) in a A/B test to collect the log data**. I also appreciate the policy code being released, as it also helps with reproducibility. The design of this benchmark is elegant.
>
> If you still feel that our claim is too strong, we will simply point out that our dataset is suitable for the comparisons of OPE estimators, as you suggested.
>
> ---
> > One minor point: why is there a time series split point for evaluation and test? Are the two policies run simultaneously? I would think that, even in the out of sample case, the full dataset could be used for the on-policy estimator.
>
> Many thanks for the question. To answer it, suppose that we aim to apply OPE to a real-world problem. Moreover, we update our decision-making policy every three days.
>
> A behavior policy $\pi_b$ collected logged bandit feedback data $D$ from Tuesday to Thursday.
> We now update and evaluate the policy using the log data and deploy the new policy from Friday to Sunday. In this situation, the evaluation set in our out-of-sample case corresponds to the logged bandit feedback data $D$ used in OPE, which was collected from Tuesday to Thursday. Instead, the new policy (evaluation policy) will be used from Friday to Sunday, and thus its performance should be measured during this time period.
>
> Assuming this realistic OPE application, we applied a time series split and used only an evaluation set to conduct OPE and a test set different from the evaluation set to measure the performance of the evaluation policy $\pi_e$.
>
> (It is true that most of the research papers assume that there does not exist any distributional shift between evaluation and test sets. We also include this setting as the in-sample case. In this case, we used the full dataset to measure the performance of the evaluation policy, as you suggested. We believe that our benchmark experiments are more comprehensive than most of the existing works with respect to this “in-sample vs out-sample” point.)

---

### Official Review · Reviewer_7mFh · 2021-07-04

**Rating:** 4
**Confidence:** 4

**Strengths:**

1. The dataset seems useful for comparing model-free OPE estimators. The size of the dataset and the inclusion of two different behavior policies makes this dataset uniquely useful for OPE research, in a way that I believe to be novel.

2. The empirical benchmark included many of the most important baselines to well understand the performance of OPE algorithms on this dataset. The results seem easily reproducible.

**Weaknesses:**

1. The user features seem to not be predictive of the reward at all, performing seemingly no better than random with the models used by this empirical study. I wonder if this is due to the feature hashing, which was used for anonymization? Generally, hash functions seek to take even highly similar inputs and map them to entirely different outputs, making generalization across similar context vectors impossible.

2. The paper makes repeated promises of *multiple* behavior policies, however only two policies are currently available. While this is sufficient for off-policy experiments and it is technically true that two policies are "multiple", it did feel a bit misleading to discover that multiple=2 on page 5.

3. The writing is highly repetitive, with key aspects of the dataset repeated throughout the paper (e.g. that there are multiple policies, the discription of Zozotown, etc.). It seems that the paper could be reduced by at least a couple of pages without harming readability, clarity, or precision. Normally, I wouldn't comment on this except I do feel there were a few omissions which could be included if the vertical space was used more sparingly.

4. One such omission: there is negligible investigation of the dataset itself. What are the empirical probabilities of selecting actions (e.g. proportion that each action is taken) for the Bernoulli policy? How do certain components of the context vector impact action selection (e.g. are features of the user important or only features of the item)?

**Additional Feedback:**

It would be useful for Table 4 to highlight best, second-best, and worst estimators based on the confidence intervals instead of using only the mean. If two confidence intervals overlap, then both estimators should have the same color (otherwise implicit claims are statistically misleading).

It would benefit the paper to discuss computational performance of the implemented OPE estimators in OBP. Are experiments easily parallelizable with the benchmark code? How strongly does the code rely on scikit-learn? Requiring a week to complete the most expensive experiment is long enough to inhibit extensive iterative testing of algorithmic developments during the research process, but perhaps this took so long due to running estimators in serial instead of in parallel?

**Clarity:**

The writing has a lot of redundancy. Removing this would enhance readability and would allow space in the main paper for either more empirical investigation of the dataset, or at least to add some of the details already in the appendix to the main body.

**Correctness:**

My only concern is the use of hashing for anonymity of the data. I agree completely with the need to anonymize, but I wonder if other methods of anonymization would be more appropriate (e.g. even simply dropping certain user features from the public data release might still allow user context vectors to be predictive, while anonymous).

---

## Edit after discussion

This point is very important to me and I believe it is a blocker for publication. The dataset is proposed for OPE research and is released as a contextual bandit dataset, but it appears that the contexts no longer retain any predictive information due to the feature hashing. This is a strong enough deviation from the current claims of the paper, that I believe it is (unintentionally) misleading and would significantly reduce community adoption of the dataset.

I have two concrete alternative suggestions.
1. Rework the paper to make this limitation more clear and investigate (a) to what extent predictability was lost from feature hashing (e.g. use the original unhashed features and compare to an estimator that only uses the hashed features, are they equally predictive?) and (b) indicate that this means that this dataset cannot be used for contextual bandit-based research.
2. Find an alternative to feature hashing. I sympathize that this is a restriction of the company, but in its current state I am unsure that the dataset provides utility for current research for OPE methods. Some alternatives would be dropping certain identifying features, projecting the feature vectors onto a lower dimensional manifold so that some information is lost, randomly perturbing feature to disrupt identifiability, etc. But similar to point (1), the impact of these decisions should be investigated within the paper.

**Documentation:**

The dataset has clear documentation, though more information about the context vectors would be useful. The benchmark code seems clear and easy to use (I did not try using it, but I did skim through the code which appears well-written and well-documented).

Maintenance and hosting appear to be in order and are discussed in the paper.

**Ethics:**

In its current form, no there is no need for further ethical review. However, I strongly prefer to change the anonymization method of user information (which currently favors adhering to ethical considerations at the cost of technical considerations); it is possible that iterating on this point would benefit from a reviewer who is more knowledgeable in effective anonymization without sacrificing predictability of the data.

**Relation To Prior Work:**

This is well done.

Because reviewers always have the luxury of asking for more, the only potential relevant omissions that I see are:
1. Comparison of with the DICE family of methods as well as the MAGIC estimator (Thomas and Brunskill, 2016)
2. Further discussion of the conclusions of this empirical benchmark using the proposed dataset in comparison with other empirical comparison using synthetic data. I recognize Section 5.3 does this in a limited way currently, but comparing more directly to a particular paper would make the conclusions more impactful (e.g. "here is a particular conclusion that our new dataset makes apparent, which is different from the conclusion of <some other paper> which used only synthetic datasets").

I believe covering these omissions would enhance the quality of the paper; however, I do not find these omissions detract at all significantly from the paper in its current form.

**Summary And Contributions:**

This paper introduces a novel dataset and code library for off-policy policy evaluation for contextual bandits. The dataset contains 26m visits to a fashion retail webpage where three items are displayed to a user, tracking user information, and number of clicks for each item. The data was collected following two different behavior policies (uniform random and a learned Bernoulli policy). The paper includes an in-depth empirical benchmark of popular OPE estimators on the proposed dataset.

---

> ### Author Response · Authors · 2021-07-14
> **Author response**
>
> Thank you for your useful, detailed feedback. We will update the paper with the suggested minor revisions and respond below to some concrete questions and comments.
>
> ---
> > Comparison of with the DICE family of methods as well as the MAGIC estimator (Thomas and Brunskill, 2016)
>
> Thank you for proposing the potential extension of our work. We agree that our package has room for improvements.
>
> We would like to clarify that we have already included Switch-DR proposed by (Wang et al., 2017) [39]. In Section 4.1 of [39], the authors state that MAGIC is similar to Switch-DR, and MAGIC focuses more on the reinforcement learning setting. We rather think that what is lacking from our package is the automatic hyperparameter tuning of the estimators’ hyperparameters, which both (Wang et al., 2017) and (Thomas and Brunskill, 2016) proposed.
>
> Therefore, following your comment, we have implemented the automatic hyperparameter tuning for Switch-DR and also DRos proposed by the original papers [29, 39]. The relevant pull request is available here:  https://github.com/st-tech/zr-obp/pull/116
>
> We will add the results of Switch-DR and DRos with a tuned hyperparameter in Table 4.
>
> We are also extending the problem setting of the package and implementing OPE estimators that can address continuous action space. The relevant pull requests are below.
> - https://github.com/st-tech/zr-obp/pull/112
> - https://github.com/st-tech/zr-obp/pull/113
> - https://github.com/st-tech/zr-obp/pull/114
>
> As stated here, we will not stop developing our package. Users of the package can easily follow the progress in our google group mailing list: https://groups.google.com/g/open-bandit-project
>
> ---
> > The user features seem to not be predictive of the reward at all, performing seemingly no better than random with the models used by this empirical study. I wonder if this is due to the feature hashing, which was used for anonymization?
>
> First of all, we followed the company’s decision and cannot avoid using feature hashing. However, we still think that this point does not lose the value of our open-source much. In web-industry applications, we often cannot accurately predict the reward variable such as conversion logs due to the sparsity of the reward and feature. There is also the delayed feedback issue [a]. We think that the performance of $\hat{q}$ reported on Table 6 is not wired from the application perspective.
>
> It is true that the performance of $\hat{q}$ greatly affects the performance of OPE estimators, and we have already acknowledged and described this as a reason to explain our benchmark results in Table 4 (model based estimators such as DM are suffering). We will add more comments as to what challenges the dataset provides in the main text (Sections 4, 5, and 6).
>
> We also expect that our open-source will inspire other researchers and companies to create new benchmarks. Then, we will have a variety of datasets such as ones with easily estimable rewards, which is a different characteristic from our data. We hope that our open-source will be a first step to build a collection of benchmark datasets that provide a variety of challenges to the community such as the Open Graph Benchmark in the graph neural network community [12].
>
> **reference**
>
> [a] Olivier Chapelle. 2014. Modeling delayed feedback in display advertising. In Proceedings of the 20th ACM SIGKDD international conference on Knowledge discovery and data mining. ACM, 1097–1105.
>
> ---
> > The paper makes repeated promises of multiple behavior policies, however only two policies are currently available. While this is sufficient for off-policy experiments and it is technically true that two policies are "multiple", it did feel a bit misleading to discover that multiple=2 on page 5.
>
> Many thanks for the good point. As we described and demonstrated throughout the paper, having multiple (at least 2) logging policies is essential to enable the evaluation of OPE experiments. Our dataset and code is the first to include at least 2 datasets/policies and enable such OPE experiments. This point differentiates our open-source from the existing ones. We emphasized this point by using the term "multiple". We have also stated that having only 2 datasets/policies as one of the limitations in Section 6.
>
> If you still feel that the term "multiple" is confusing, we will change "multiple" to "two" in the future revision (we will at least clarify our motivation to use the term multiple).

---

> > ### Author Response · Authors · 2021-07-14
> > **Additional author response**
> >
> > ---
> > > It would benefit the paper to discuss computational performance of the implemented OPE estimators in OBP. Are experiments easily parallelizable with the benchmark code? How strongly does the code rely on scikit-learn? Requiring a week to complete the most expensive experiment is long enough to inhibit extensive iterative testing of algorithmic developments during the research process, but perhaps this took so long due to running estimators in serial instead of in parallel?
> >
> > The benchmark code has already been parallelized as follows.
> > - OPE: https://github.com/st-tech/zr-obp/blob/6d2824b1bad3f0c9e9d21fccea6d50f5e85b8357/benchmark/ope/benchmark_off_policy_estimators.py#L209-L212
> > - $\hat{q}$ training: https://github.com/st-tech/zr-obp/blob/6d2824b1bad3f0c9e9d21fccea6d50f5e85b8357/benchmark/ope/train_regression_model.py#L240-L243
> >
> > In fact, OPE itself is not computationally intensive. What is computationally challenging is to train $\hat{q}$ on  a large dataset. Specifically, when we train RandomForest on the largest dataset ($\pi_b$ is BernoulliTS and “All” Campaign) 30 times, it takes about a week. Moreover, for this largest data, the memory issue arises and we could not fully parallelize the experiment for that case (as we ran the experiments on a laptop). However, this issue can be solved by using better computational resources and fully utilizing the power of parallelization.
> >
> > Currently, we just state the time needed to run the most computationally intensive experiment. However, we can run most of the experiments within a day even with the limited computational resources, as we have already parallelized the experimental workflow. We will make an additional table in the appendix to summarize the computational time needed to run each part of the experiment.
> >
> > ---
> > > One such omission: there is negligible investigation of the dataset itself. What are the empirical probabilities of selecting actions (e.g. proportion that each action is taken) for the Bernoulli policy? How do certain components of the context vector impact action selection (e.g. are features of the user important or only features of the item)?
> >
> > Thank you for the great point. We will add some additional descriptive analysis of the dataset in the revision.

---

> > ### Comment · Reviewer_7mFh · 2021-07-14
> > **Discussion**
> >
> > I appreciate the discussion about Switch-DR and MAGIC, I had not previously realized the similarity between these estimators!
> >
> > ---
> >
> > I will admit that I remain rather concerned about the feature hashing and the predictability of the contexts. If the context vectors no longer contain any predictive value (note I am claiming that all predictability/information has been destroyed; which I believe is the case), then this dataset effectively is _not_ a contextual bandit dataset. The contexts cannot be used for any model-based research. If the company has a hard rule about feature hashing, then I think the only recompense is to rework the paper to:
> > 1. Remove all model-based estimators from the discussion
> > 2. Remove the contexts entirely from the dataset, no need to use bandwidth to download those if they contain no information.
> > 3. Be exclusively about model-free approaches to OPE
> >
> > I do not think that the lack of predictability is representative of realistic datasets, because I believe the predictability is something that was lost only during the open-source process. For instance, I believe this lack of predictability means that we can no longer reconstruct the Bernoulli-TS policy that was used for data collection. This feature hashing is not something that needs to be overcome in general realistic settings.
> >
> > To be clear, I do not believe that adding more discussion of the limitations to the paper is sufficient in this case. I believe that without predictive context vectors, the dataset is quite limited in utility for current OPE research.

---

### Decision · Program_Chairs · 2021-07-26

**Decision:**

Reject

**Comment:**

The main strength of this paper is that having two policies collecting the logged data (with recorded action probabilities) dramatically simplifies the OPE benchmarking. However, the main weakness is that the feature-hashing anonymization method is removing a majority of the information contained within the context vectors.  Ultimately, the feature-hashing anonymization seemed like a severe limitation for the usefulness of the proposed benchmark.